# Physical Performance and Non-Esterified Fatty Acids in Men and Women after Transcatheter Aortic Valve Implantation (TAVI)

**DOI:** 10.3390/nu14010203

**Published:** 2022-01-02

**Authors:** Michaela Härdrich, Anja Haase-Fielitz, Jens Fielitz, Michael Boschmann, Olga Pivovarova-Ramich, Andreas F. H. Pfeiffer, Natalia Rudovich, Karsten H. Weylandt, Christian Butter

**Affiliations:** 1Department of Cardiology, Heart Centre Brandenburg Bernau, Faculty of Health Sciences Brandenburg, Brandenburg Medical School (MHB) Theodor Fontane, 16321 Bernau, Germany; michaela.haerdrich@immanuelalbertinen.de (M.H.); christian.butter@immanuelalbertinen.de (C.B.); 2Institute of Social Medicine & Health Care Systems Research, Otto von Guericke University Magdeburg, 39120 Magdeburg, Germany; 3DZHK (German Centre for Cardiovascular Research), Partner Site Greifswald, 17489 Greifswald, Germany; Jens.Fielitz@med.uni-greifswald.de; 4Department of Internal Medicine B, Cardiology, University Medicine Greifswald, 17489 Greifswald, Germany; 5Experimental & Clinical Research Centre (ECRC), a Joint Cooperation between Charité—University Medicine Berlin and Max Delbrück Centre (MDC) for Molecular Medicine in the Helmholtz Association, 13125 Berlin, Germany; michael.boschmann@charite.de; 6Research Group Molecular Nutritional Medicine, Department of Molecular Toxicology, German Institute of Human Nutrition Potsdam-Rehbruecke, 14558 Nuthetal, Germany; Olga.Ramich@dife.de; 7Department Endocrinology and Metabolism, Charité—Universitätsmedizin Berlin, 10117 Berlin, Germany; andreas.pfeiffer@charite.de; 8German Center for Diabetes Research (Deutsches Zentrum Für Diabetesforschung e.V.), 85764 Neuherberg, Germany; 9Department of Internal Medicine, Spital STS AG, University of Zurich, 8006 Zurich, Switzerland; Natalia.Rudovich@spitalstsag.ch; 10Department of Internal Medicine, Spital Bülach, 8180 Bülach, Switzerland; 11Medical Department, Divisions of Hepatology, Gastroenterology, Oncology, Haematology, Palliative Care, Endocrinology and Diabetes, Ruppiner Kliniken, Brandenburg Medical School, 16816 Neuruppin, Germany; karsten.weylandt@mhb-fontane.de

**Keywords:** heart failure, body composition, six-minute walking test (6-MWT), coronary heart disease, diabetes mellitus, chronic kidney disease, sex-specific differences

## Abstract

Background: Men and women with valvular heart disease have different risk profiles for clinical endpoints. Non-esterified fatty acids (NEFA) are possibly involved in cardio-metabolic disease. However, it is unclear whether NEFA concentrations are associated with physical performance in patients undergoing transcatheter aortic valve implantation (TAVI) and whether there are sex-specific effects. Methods: To test the hypothesis that NEFA concentration is associated with sex-specific physical performance, we prospectively analysed data from one hundred adult patients undergoing TAVI. NEFA concentrations, physical performance and anthropometric parameters were measured before and 6 and 12 months after TAVI. Physical performance was determined by a six-minute walking test (6-MWT) and self-reported weekly bicycle riding time. Results: Before TAVI, NEFA concentrations were higher in patients (44 women, 56 men) compared to the normal population. Median NEFA concentrations at 6 and 12 months after TAVI were within the reference range reported in the normal population in men but not women. Men but not women presented with an increased performance in the 6-MWT over time (*p =* 0.026, *p =* 0.142, respectively). Additionally, men showed an increased ability to ride a bicycle after TAVI compared to before TAVI (*p =* 0.034). NEFA concentrations before TAVI correlated with the 6-MWT before TAVI in women (Spearman’s rho −0.552; *p =* 0.001) but not in men (Spearman’s rho −0.007; *p =* 0.964). No association was found between NEFA concentrations and physical performance 6 and 12 months after TAVI. Conclusions: NEFA concentrations improved into the reference range in men but not women after TAVI. Men but not women have an increased physical performance after TAVI. No association between NEFA and physical performance was observed in men and women after TAVI.

## 1. Introduction

Transcatheter aortic valve implantation (TAVI) is a standard treatment for aortic stenosis in patients with moderate to high surgical risk. Aortic valve stenosis is characterized by increased inflammation, fibrosis, and calcification of the aortic valve leaflets [1,2]. Obesity and impaired lipid metabolism have emerged as risk factors for aortic stenosis [3,4,5]. In patients with aortic stenosis, sex-related differences in clinical presentation, metabolism and pathophysiology of valvular calcification have been described [6,7]. Women seem to have less valvular calcification, but more fibrosis compared with men [8]. Additionally, recent studies have suggested sex-related differences in cardiac remodelling and reverse remodelling after TAVI [9].

Non-esterified fatty acids (NEFA) are formed during the lipolysis of adipose tissue and serve as energy suppliers predominantly metabolized through β-oxidation [10]. NEFA have proinflammatory effects and lead to insulin resistance and the inhibition of glycolysis during ischemia reperfusion [11]. In addition to metabolic effects, neuro-humoral activation has been observed in patients with heart failure and increased NEFA concentrations [12].

A reactive hyperadrenergic state and stimulation of lipolysis by catecholamines as well as elevated natriuretic peptides are contributing factors [10,12,13]. Additionally, increased NEFA concentrations impair nitric oxide-dependent and -independent vasodilatation and contribute to the formation of oxygen radicals and endothelial dysfunction [14]. Possible consequences are increased blood pressure and the formation of atherosclerotic plaques due to NEFA accumulation in blood vessels [15,16]. Female patients with aortic stenosis tend to walk for shorter time periods on a treadmill and achieve lower metabolic equivalents [17].

In sum, men and women with valvular heart disease have different risk profiles for clinical endpoints. As NEFA have been shown to be involved in cardio-metabolic disease and might increase cardiometabolic risk, they might play a role as biomarkers in this context. However, whether NEFA concentrations differ according to sex and if their levels are associated with physical performance after TAVI has not yet been reported.

Accordingly, we evaluated sex-specific concentrations of NEFA before and after TAVI and analysed if there was a sex-specific relationship of NEFA with physical performance as measured by a six-minute walking test (6-MWT) and weekly bicycle riding time.

## 2. Materials and Methods

### 2.1. Study Design and Patients

This prospective single-centre cohort study was conducted at Brandenburg Heart Centre during January 2017 and July 2019. Within this time, 1264 TAVI procedures were performed in our department. The study was approved by the local ethics committee AS 87(bB)/2015. Written informed consent to participate in the study and to publish the results was obtained from all individual patients included in the study.

Inclusion criteria comprised symptomatic heart failure, severe aortic stenosis with planned TAVI and written informed consent. Exclusion criteria were lack of legal capacity, life expectancy below 1 year and missing baseline blood draw.

### 2.2. Anthropometric Analysis

Body weight was measured in light clothing and without shoes. Waist size was measured at the shortest point below the lower rib margin and the iliac crest. Body mass index (BMI) was calculated as weight (kilograms) divided by height squared (meters).

Measurement of skinfold thickness was used to indirectly determine total body fat, adjusting for age and sex. Skinfold thickness was measured at three different sites (triceps, back, hips) using a calliper, i.e., fat-measuring forceps. On the triceps, a skinfold was lifted parallel to the long axis of the upper arm and measured on the dorsal side of the upper arm over the triceps’ brachii muscle midway between the acromion and olecranon. With the arm hanging, a skinfold was then determined on the dorsum just below the inferior tip of the scapula. The skinfold was oblique to the long axis of the body. Last, a measurement was taken on the lateral axillary line just above the iliac crest. The skinfold was oriented parallel to the long axis of the body. The three results were added together and expressed in millimetres. Measurements were carried out by the same operator; fat mass was collected and analysed by specific formulas provided by the manufacturer.

### 2.3. Physical Performance

Six-minute walking test (6-MWT) before and after TAVI was used as sub-maximal exercise test to assess aerobic capacity and endurance. Patients were asked to walk as far as possible within 6 min. The distance was measured with a distance measuring wheel. After completion of the 6 min test or after the patient had stopped, the Borg scale was assessed. Patients were asked to rate their exertion on the scale during the activity, combining all sensations and feelings of physical stress and fatigue. The scale ranges from 6—corresponding to an activity that is not strenuous at all—to 20, which is perceived by the patient as maximally strenuous. In addition, patients were asked for physical activity records measured as time spent bicycle riding weekly before and after TAVI.

### 2.4. Sample Collection

Blood samples were taken one day after hospital admission from the fasting patient for subsequent biomarker determination. Serum and Ethylenediaminetetraacetic acid (EDTA) plasma was prepared by centrifugation at 1452 g and 5 °C for 10 min and were frozen at −80 °C till analyses. NEFA were assessed in serum samples with Pentra C400 benchtop analyser (Horiba Medical, Grabels, France) using NEFA-HR (2) Assay (FUJIFILM Wako Chemicals Europe GmbH, Neuss, Germany) according to the manufacturer’s instructions.

In addition, on the day of hospital admission, the following laboratory parameters were routinely determined in all patients: blood count, electrolytes, renal retention parameters (estimated glomerular filtration rate, eGFR according to Chronic Kidney Disease, CKD-EPI formula, serum creatinine), lipid status (total cholesterol, LDL and HDL cholesterol, triglycerides), C-reactive protein (CRP), NT-Pro-BNP, blood glucose and HbA1c. For routine blood measurements the autoanalyzer COBAS Integra 800 (Roche Diagnostics, Basel, Switzerland) was used.

### 2.5. Data Collection

A 12-lead Electrocardiogram (ECG) was recorded prior to TAVI and post TAVI. Demographic variables, comorbidities, medications, New-York Heart-association (NYHA) classification and logistic EuroSCORE (European System for Cardiac Operative Risk Evaluation) were collected.

Post TAVI, type and size of valve were recorded as well as any complications occurring peri or post TAVI. Relevant complications were bleeding, occurrence of vascular complications after femoral puncture, respiratory insufficiency, the development of delirium or acute kidney injury, stroke, occurrence of a new left bundle branch block and a necessary device implantation (pacemaker, ICD, or CRT). Length of stay in hospital or the intensive care unit as well as patient’s discharge status (discharge to home, direct rehabilitation, transfer to inpatient care at another hospital) were recorded. Finally, mortality during hospitalization and after 6 and 12 months, as well as possible rehospitalization within 30 days of hospital discharge, were recorded.

### 2.6. Transthoracic Echocardiography

All patients underwent transthoracic echocardiography preoperatively to quantify aortic valve stenosis. Both mean and maximum pressure gradients across the aortic valve were determined by continuous-wave Doppler. When the mean pressure gradient exceeded 40 mmHg, aortic valve stenosis was considered high-gradient aortic valve stenosis. When the mean pressure gradient was less than 40 mmHg and the valve orifice area determined by the continuity equation was <1.0 cm², a diagnosis of low-flow low-gradient aortic valve stenosis with preserved LVEF was made if the LVEF was greater than 50% and the stroke volume index was <35 mL/m². If these criteria were present at an LVEF less than 50%, it was classified as high-grade aortic valve stenosis with reduced ejection fraction. In unclear cases, transoesophageal echocardiography was performed to confirm the diagnosis by planimetric determination of the valve orifice area. LVEF was determined via biplane according to Simpson’s biplane method in 4-chamber and 2-chamber views; in the few cases where the acoustic conditions were insufficient, it had to be estimated visually. The classification of normal, low-grade, moderate-grade and high-grade LVEF followed the Manual for the Indication and Performance of Echocardiography—Update 2020 of the German Society of Cardiology.

### 2.7. TAVI Procedure

Transcatheter aortic valve implantation has been performed at the Brandenburg Heart Centre since 2008 and serves as an alternative, minimally invasive procedure to conventional aortic valve replacement in patients with aortic valve stenosis and high surgical risk. The diagnosis is confirmed by transthoracic echocardiography and, if necessary, by transoesophageal echocardiography in case of ambiguity. If the procedure has not already been performed by the referring hospital, each patient receives a left heart catheter examination to determine the current coronary status and, if necessary, therapy for stenosis requiring treatment. A computer tomography (CT)-scan of the heart and the pelvic–leg vasculature also provides important information for surgical planning. At the Brandenburg Heart Centre, the transfemoral approach is used in most cases. The pelvic–leg vessels are measured, and the thoracic aorta is assessed. Annulus as well as the distance to the coronaries are determined in order to select the most suitable valve type preoperatively. When all these data are available, the heart team, consisting of cardiologists, cardiac surgeons, and anesthesiologists, will make the indication for further therapy. TAVI is predominantly performed under general anaesthesia. In some cases, especially when a high risk of delirium and no possible peri-interventional complications are anticipated, TAVI is also performed under analgo-sedation. Depending on valve anatomy and calcification, a predilatation may be required. In the majority a direct implantation is performed. Under angiographic control, the valve prosthesis is positioned and finally released under tachycardic pacing, which is induced by a passaged pacemaker that has been washed in beforehand. Finally, the femoral placed sheaths are removed, and the patient is transferred to cardiac intermediate care via the recovery room with the femoral pressure dressing in place.

Follow-up care was performed by the heart failure outpatient clinic of the Brandenburg Heart Centre after 6 and 12 months and comprised transthoracic echocardiography, laboratory tests, 12-lead ECG, and a physical examination as well as a questionnaire including information regarding weekly duration of exercise.

### 2.8. Statistics

Statistical analysis plan was approved by the authors before analyses began. As this was a pilot study, we used a convenient sample size for testing the reliability and validity of the data collected.

Categorical data were reported as percentages with a 95% confidence interval (CI) of the mean percentage and compared using the Fisher exact test. After testing for normal distribution, continuous data were reported as the median with 25th to 75th percentiles, and nonparametric data were compared using the Mann–Whitney U test. The Friedmann test was used to evaluate changes over time within one group. We used nonparametric bivariate correlation and report Spearman correlation coefficients. We used list-wise deletion when generating boxplots, including several endpoint measurements over time and pair-wise deletion for baseline values presented in the tables.

The ability of NEFA to predict (i) improvement in NYHA class, (ii) improvement in the distance of the 6-MWT and (iii) rehospitalisation within 30 days was assessed by plotting receiver operating characteristic (ROC) curves and further reported as area under the curve (AUC) with 95% CIs. An AUC-ROC value of >0.7 was taken to indicate a reasonable performance [18]. Information on missing data is provided in the tables’ footnotes.

Statistical significance is denoted by 2-sided *p*-values < 0.05. Statistical analysis was performed using SPSS 26.0 (SPSS Inc., Chicago, IL, USA).

## 3. Results

### 3.1. Patient Characteristics

We analysed data from one hundred adult patients undergoing TAVI in our institution. A CONSORT diagram for the study is shown in Figure 1. The median age was 82 years (25–75th percentile 78–85). Forty-four percent of patients were women. Most patients presented with heart failure with preserved ejection fraction (72%) and NYHA class III (71.9%). Sixty-eight percent of all patients had chronic kidney disease with a median estimated glomerular filtration rate (eGFR) before TAVI of 54 mL/min/1.73 m^2^ (25–75th percentile 41–71). The median logistic EuroSCORE was 12.8% (25–75th percentile 8.4–19.5). The full patient baseline characteristics overall, separated by sex, are shown in Table 1.

A higher proportion of women presented with a BMI outside the age-adjusted reference range compared to men (68.2% vs. 41.1%, *p =* 0.007). Total body fat before TAVI was higher in women vs. men (median 30.4%, 25–75th percentile 27.3–34.3 vs. 22.0%, 25–75th percentile 17.6–25.2, *p* < 0.001). The logistic EuroSCORE, medication at hospital admission and comorbidities did not differ between women and men, except for the prevalence of coronary heart disease, which was higher in men vs. women (80.4% vs. 56.8%, *p =* 0.011) (Table 1).

### 3.2. NEFA Concentrations before and after TAVI

Before TAVI, NEFA concentrations were higher in the patients compared to the normal population [19,20] (Figure 2). Overall, median NEFA concentration decreased from baseline (0.68 mmol/L [0.44–0.86]) to 6 months (0.40 mmol/L [0.31–0.60]) and 12 months post TAVI (0.49 mmol/L [0.20–0.60]), *p <* 0.001. Sex-specific changes in NEFA over time are shown in Figure 2. Median NEFA concentrations at 6 and 12 months after TAVI were within the reference range reported in the normal population in male but not in female patients.

### 3.3. Clinical Outcomes

A total of 55.7% of all patients showed an improvement in the 6-MWT. The NYHA class improved in 66.2% of patients over time (Table 2). Overall, the median length of stay in hospital was 13 days (9–17). Perioperative vascular complication occurred in four patients (all male). The rehospitalisation rate within 30 days was 10.2%, and the 12-month mortality was 6%. Hospital outcome and mortality did not differ between men and women (Table 2).

NEFA concentrations before TAVI did not predict improvement in NYHA class (AUC-ROC 0.460 [95% CI 0.316–0.605], *p =* 0.586), improvement in the distance of the 6-MWT (AUC-ROC 0.571 [95% CI 0.423–0.718], *p =* 0.349) or rehospitalisation within 30 days (AUC-ROC 0.633 [95% CI 0.461–0.805], *p =* 0.191) with no sex-specific effect observed (all AUC-ROC < 0.68, all *p >* 0.15).

### 3.4. Physical Performance before and after TAVI

Men but not women presented with an increased performance in 6-MWT over time (*p =* 0.026, *p =* 0.142, respectively), Figure 3a. Additionally, according to the weekly duration of exercise, men showed an increased ability to ride a bicycle after TAVI compared to before TAVI (*p =* 0.034). There were no changes in bicycle riding in female patients over time (Figure 3b).

### 3.5. Relationship of NEFA Concentrations with Physical Performance

NEFA concentrations before TAVI correlated with the 6-MWT before TAVI in female (left panel) (Spearman’s rho −0.552; *p =* 0.001) but not in male (right panel) (Spearman’s rho −0.007; *p =* 0.964), Figure 4.

Separated by sex, there was no correlation of NEFA concentrations at 6 and 12 months with 6-MWT or bicycle riding at 6 and 12 months (Table 3).

### 3.6. Relationship of NEFA Concentrations with Laboratory and Echocardiographic Parameters

In the total patient cohort, we observed a weak correlation of NEFA concentration before TAVI with body mass index (BMI; Spearman’s rho 0.230), fasting glucose (Spearman’s rho 0.240), CRP (Spearman’s rho 0.281) and IL-6 (Spearman’s rho 0.235). Separated by sex, NEFA concentration before TAVI did not correlate with age (Spearman’s rho −0.058), NT-proBNP concentration before TAVI (Spearman’s rho 0.183) or baseline eGFR (Spearman’s rho −0.070). NEFA concentrations before TAVI did not correlate with lipid parameters including cholesterol, LDL, HDL, and triglyceride before TAVI, skinfold thickness, waist size and total body fat (all Spearman’s rho <0.15). Additionally, there was no correlation of NEFA with any echocardiographic parameters of either systolic or diastolic function measured before TAVI.

## 4. Discussion

In a prospective cohort study of 100 patients undergoing TAVI, we assessed NEFA as a biomarker indicating cardiometabolic risk and its sex-specific association with physical performance before and 6 and 12 months after intervention. The study participants represented a typical patient cohort with aortic stenosis undergoing TAVI.

Here, we report that in both men and women, NEFA concentrations before TAVI were above those of elderly cardiac, non-TAVI patients. Importantly, NEFA concentrations improved into the normal range in men but not women after TAVI. Before TAVI, NEFA concentrations inversely correlated with distance in the 6-MWT in women but not in men. We also found sex-specific differences in the physical performance of patients after TAVI and showed a significant improvement over 12 months only in men. Finally, we observed a decrease in NT-proBNP after TAVI in men and women.

Under normal conditions, the healthy heart derives two-thirds of its energy from free fatty acids [21]. Elevated concentrations of NEFA may lead to increased myocardial uptake, increased triglyceride synthesis and fat storage within cardiomyocytes, resulting in lipotoxicity, apoptosis and left ventricular dysfunction [12,13,14,22]. The Cardiovascular Health Study found that NEFA concentrations were associated with a higher risk of heart failure in older adults [23]. Additionally, elevated NEFA concentrations were associated with increased 3-month mortality in patients with acute heart failure; however, sex-specific effects have not been reported [13].

Interestingly, in human studies, ANP has stimulated lipid mobilization, modulated insulin secretion, and inhibited lipolysis [24,25]. Circulating levels of natriuretic peptides increase in accordance with the severity of the heart failure [26]. Thus, elevated NEFA concentrations in heart failure circumstances are not only depended on the activation of the sympathoadrenergic system but also activate the natriuretic peptide system. Sex differences in visceral fat lipolysis are well documented [27]. It is known that, especially in visceral fat, lipolysis is more active in men and is also more rapidly downregulated by the decrease in lipolytic factors such as BNP [28]. In this regard, lipolysis induced by agents acting at the adenylate cyclase and protein kinase A levels were almost enhanced two-fold in men [29].

There may be several potential explanations for the study findings. The missing correlation of NEFA with NT-proBNP levels, which were not significant before or after TAVI and lack of sex-specific changes after TAVI suggest that sex-specific differences in NEFA may not be explained by a direct link to natriuretic peptides in the present study, a link which has been established for men in chronic heart failure and after ANP infusion [30,31].

In the present study, NEFA concentrations over time appear to reflect an increased metabolism stimulated by physical performance in men but not in women. According to the Randle cycle [32], decreased NEFA plasma levels result in an improved glucose uptake into skeletal muscle, which might explain the observed improved physical performance in men six months after TAVI. The observed inverse correlation between NEFA and the 6-MWT distance at baseline in women but not men might be related to a higher BMI, a slightly higher age and more severe heart failure with higher proportion of NYHA class IV in women compared to men. In the present study, men had a lower total body fat compared with women. They were able to walk a farther distance in 6-MWT before TAVI. We speculate that with different fat distribution and higher NEFA levels, men had better exercise status before TAVI.

Physical performance can be trained in cardiac insufficiency and leads to better oxygen extraction despite similar cardiopulmonary function. The observed sex-specific differences in changes in NEFA concentration over time could be due to higher motivation in men to demonstrate an improved physical performance after TAVI. Contrarily, female sex was recently reported to be an independent factor for reduced exercise capacity improvement after TAVI measured by 6-MWT [33].

NEFA concentrations reported in more than 10,000 partly elderly patients with different cardiovascular diseases [34,35,36] ranged within the upper limit of the reference range reported for normal population [19,20]. However, for patients undergoing TAVI with aortic valvular stenosis in the present study, we found NEFA concentrations above the upper limit of the reference range of the normal population. Reasons for this observation might be an increased cardiometabolic risk profile along with increased age, increased BMI or the decreased heart function of patients undergoing TAVI.

Our study has several strengths and limitations. We investigated a relatively large patient cohort followed over a relatively long time reporting metabolic and anthropometric parameters and physical performance indices. However, the study was performed at one centre, limiting the generalizability of the study findings. Physical performance was in part self-reported; however, the findings of bicycle riding were confirmed by the 6-MWT. We did not measure ANP plasma levels in this study but provided NT-proBNP plasma levels. Additionally, we did not obtain data on frailty as a potentially critical determinant of functional change after TAVI.

We did not assess preexisting orthopedic or psychiatric diseases such as depression in the study; both comorbidities may be more prevalent in elderly women and could affect performance of the 6-MWT. Furthermore, given that NEFAs are composed of a complex set of different fatty acids, a limitation of our study is that we only measured the total content of NEFA, and not individual fatty acid species contained in this lipid fraction.

Our findings are novel regarding the kinetics of NEFA during 12 months after TAVI and their sex-specific changes. The findings may imply that metabolic profile and physical performance are different in men and women. Such observations may be considered in risk assessment and post TAVI rehabilitation programs if study findings can be confirmed.

Future studies systematically implementing exercise capacity assessment pre- and post TAVI might help to improve patient risk stratification and evaluate the potential sex-specific role of NEFA.

## 5. Conclusions

In this hypothesis-generating study, NEFA concentrations appear to improve in men but not women into the reference range after TAVI. This was associated with an increased physical performance in men compared to women. Corresponding mechanisms uncovering potential sex-related cardiometabolic differences need to be explored in further studies.

## Figures and Tables

**Figure 1 nutrients-14-00203-f001:**
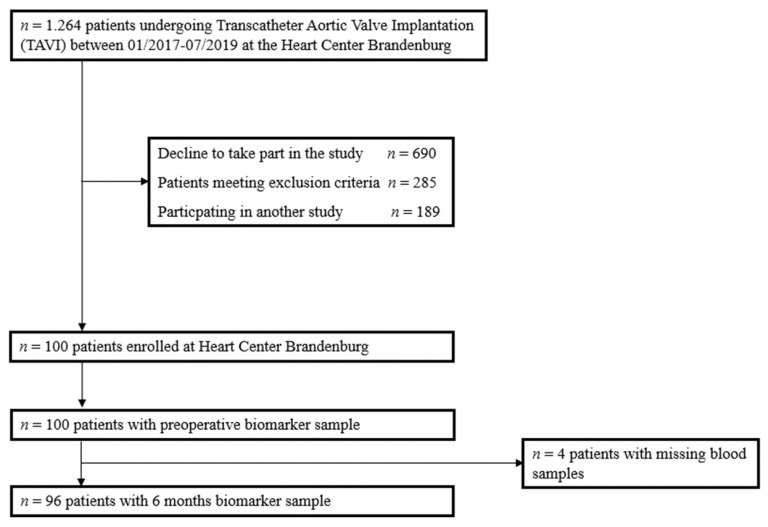
Patient flow through the study.

**Figure 2 nutrients-14-00203-f002:**
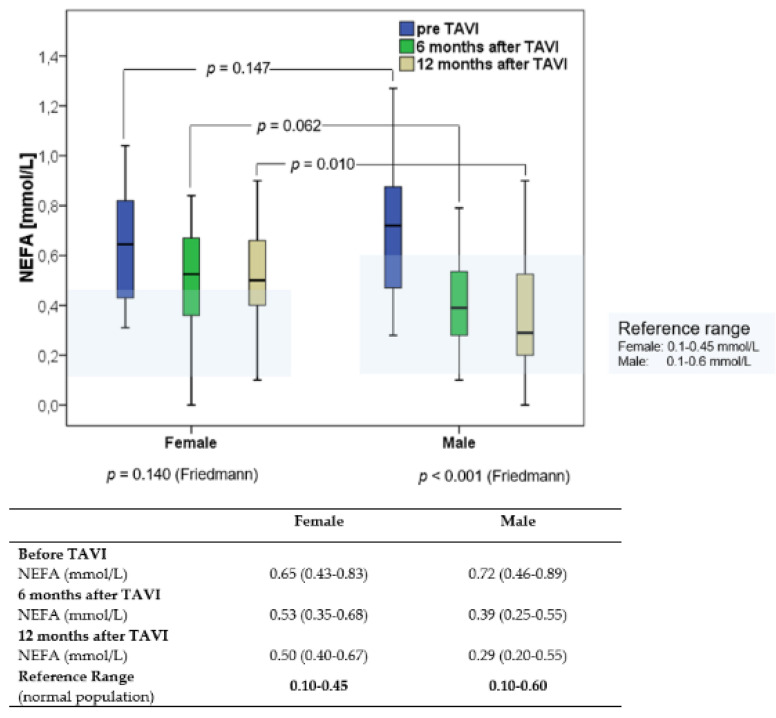
Sex-specific changes in non-esterified fatty acids (NEFA) over 12 months after Transcatheter Aortic Valve Implantation.

**Figure 3 nutrients-14-00203-f003:**
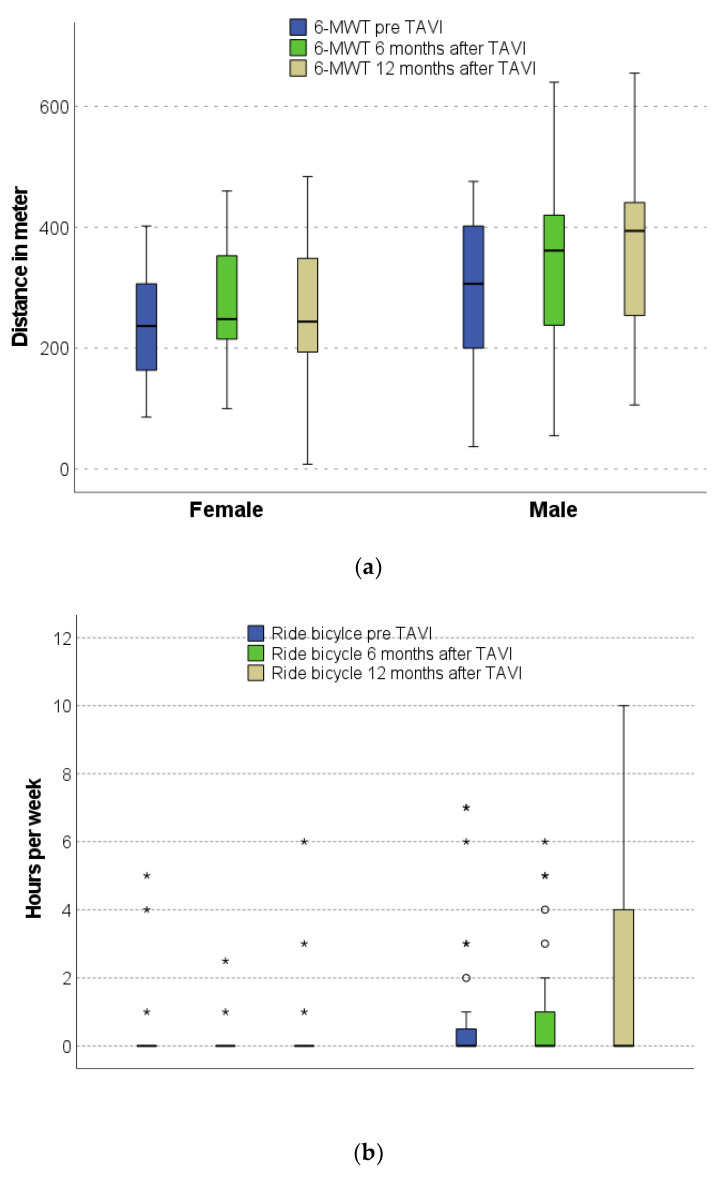
Sex-specific changes in physical performance over 12 months: (**a**) Six-minute walking test (6-MWT); (**b**) bicycle riding. The * means outliers.

**Figure 4 nutrients-14-00203-f004:**
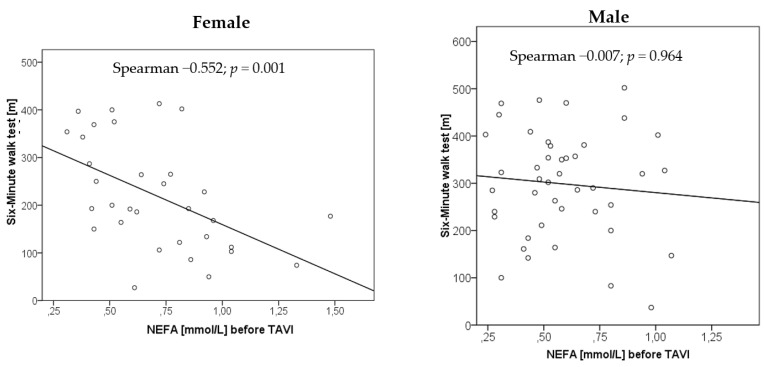
Association between NEFA concentration before TAVI and distance in six-minute walk test.

**Table 1 nutrients-14-00203-t001:** Baseline characteristics and comorbidities.

Demographics				
	Overall Cohort *n =* 100	Female *n =* 44	Male *n =* 56	*p*-Value
Age (years)	82.0 (78.0–85.0)	82.5 (78.3–84.8)	81.0 (77.3–85.0)	0.808
Blood pressure, systolic (mmHg)	125 (111–135)	125 (115–136)	122 (110–135)	0.394
Blood pressure, diastolic (mmHg)	70 (64–77)	73 (66–86)	70 (61–78)	0.284
BMI (kg/m^2^)	27.7 (24.7–30.9)	28.5 (23.7–33.5)	27.2 (25.3–30.4)	0.697
BMI > 29, *n* (%)	36 (36%)	19 (43.2%)	17 (30.4%)	0.185
BMI 24–29, *n* (%)	47 (47%)	14 (31.8%)	33 (58.9%)	0.007
BMI < 24, *n* (%)	17 (17%)	11 (25.0%)	6 (10.7%)	0.059
Skinfold thickness (mm)	52 (38–66)	54 (45–76)	50 (35–64)	0.270
Waist size (cm)	106 (99–113)	106 (94–113)	106 (100–114)	0.312
Total body fat (%)	24.6 (19.2–29.3)	30.4 (27.3–34.2)	22.0 (17.6–25.2)	<0.001
**Level of functional exercise capacity**				
Distance in 6-MWT (m)	264 (173–356)	197 (131–330)	309 (229–381)	0.007
Borg scale	13 (12–15)	13 (12–15)	13 (12–15)	0.505
**Cardiac function**				
LVEF (%)	56.0 (48.0–60.0)	55.5 (50.8–60.0)	56.0 (45.0–60.0)	0.569
NT-proBNP (pg/mL)	1.639 (734–4.841)	1.867 (772–6.258)	1.382 (583–3.910)	0.266
Haemoglobin (mmol/L)	7.8 (7.1–8.6)	7.4 (7.0–8.0)	8.1 (7.2–8.9)	0.007
Haemoglobin < Sex-specific Cut-off *, *n* (%)		23 (52.3%)	33 (59.0%)	0.506
Albumin (g/L)	36.9 (34.7–39.4)	36.9 (35.0–40.2)	37.1 (34.1–39.1)	0.635
**Heart failure**				
HFrEF, *n* (%)	13 (13%)	5 (11.4%)	8 (14.3%)	0.150
HFmrEF, *n* (%)	13 (13%)	3 (6.8%)	10 (17.9%)	
HFpEF, *n* (%)	72 (72%)	34 (77.3%)	38 (67.9%)	
**NYHA class ^~^**				
NYHA II, *n* (%)	15 (15%)	3 (7.1%)	12 (22.2%)	0.205
NYHA III, *n* (%)	69 (69%)	31 (73.8%)	38 (70.4%)	
NYHA IV, *n* (%)	11 (11%)	7 (16.7%)	4 (7.4%)	
**Comorbidities**				
Logistic EuroSCORE (%)	12.8 (8.4–19.5)	12.8 (9.8–19.0)	12.4 (7.4–20.0)	0.346
Arterial Hypertension, *n* (%)	94 (94.0%)	41 (93.2%)	53 (94.6%)	0.760
Chronic kidney disease ^#^, *n* (%)	68 (68.0%)	34 (77.3%)	34 (60.7%)	0.078
eGFR (mL/min/1.73 m^2^)	54 (41–71)	52 (39–67)	55 (41–77)	0.238
Diabetes mellitus, *n* (%) (oral medication)	24 (24.0%)	8 (18.2%)	16 (28.6%)	0.227
Insulin-dependent diabetes, *n* (%)	22 (22.0%)	8 (18.2%)	14 (25%)	0.414
HbA1c (%)	5.9 (5.6–6.3)	5.9 (5.4–6.2)	5.9 (5.6–6.4)	0.272
Chronic obstructive pulmonary disease, *n* (%)	20 (20.0%)	7 (15.9%)	13 (23.2%)	0.365
Coronary heart disease, *n* (%)	70 (70.0%)	25 (56.8%)	45 (80.4%)	0.011
Hyperlipidaemia, *n* (%)	67 (67.0%)	26 (59.1%)	41 (73.2%)	0.136
Peripheral vascular disease, *n* (%)	25 (25.0%)	10 (22.7%)	15 (26.8%)	0.642
Atrial fibrillation, *n* (%)	55 (55.0%)	23 (52.3%)	32 (57.1%)	0.688
Coronary artery bypass grafting, *n* (%)	14 (14.0%)	2 (4.6%)	12 (21.4%)	0.016
Previous myocardial infarction, *n* (%)	17 (17.0%)	7 (15.9%)	10 (17.9%)	>0.99
Previous stroke, *n* (%)	15 (15.0%)	5 (11.4%)	10 (17.9%)	0.412
Malignant disease, *n* (%)	21 (21.0%)	12 (20.5%)	9 (16.1%)	0.172
**Medication**				
ACE inhibitors, *n* (%)	43 (43.4%)	16 (37.2%)	27 (48.2%)	0.274
AT1 blockers, *n* (%)	32 (32.3%)	14 (32.6%)	18 (32.1%)	0.965
Aldosterone antagonists, *n* (%)	20 (20.2%)	9 (20.9%)	11 (19.6%)	0.874
Betablocker, *n* (%)	76 (76.8%)	36 (83.7%)	40 (71.4%)	0.151
Loop diuretics, *n* (%)	63 (63.6%)	29 (67.4%)	34 (60.7%)	0.490
Thiazide, *n* (%)	25 (25.3%)	15 (34.9%)	10 (17.9%)	0.053
Statins, *n* (%)	73 (73.7%)	32 (74.4%)	41 (73.2%)	0.893
Ezetimibe, *n* (%)	4 (4.0%)	2 (4.7%)	2 (3.6%)	0.787
DOACs, *n* (%)	30 (30.3%)	13 (30.2%)	17 (30.4%)	>0.99

6-MWT, six-minute-walking test; NT-pro BNP, natriuretic peptide, HFrEF, heart failure with reduced ejection fraction; HFpEF, heart failure with preserved ejection fraction; HFmEF, heart failure with mid-range ejection fraction; ACE, Angiotensin-converting enzyme; AT1, angiotensin-1; BMI, body mass index; LVEF, left ventricular ejection fraction; DOAC, direct oral anticoagulants; ^#^ chronic kidney disease defined as eGFR < 60 mL/min/1.73 m^2^; * female: 7.5–9.9 mmol/L; male: 8.4–10.9 mmol/L; ^~^ *n* = 5 patients with NYHA class 1; using pair-wise deletion (for missing values).

**Table 2 nutrients-14-00203-t002:** Outcome data.

Outcome				
	Overall Cohort *n =* 100	Female *n =* 44	Male *n =* 56	*p*-Value
∆ Body mass index (kg/m^2^) 6 months after TAVI	−0.31 (−0.72–0.69)	−0.32 (−0.75–0.42)	−0.30 (−0.63–0.84)	0.383
NT-proBNP (pg/mL) 6 months after TAVI	1.053 (379–1.935)	1.320 (518–2.367)	878 (324–1.33)	0.185
Improvement in 6-MWT >10%, *n* (%)	34 (55.7%)	14 (60.9%)	20 (52.6%)	0.530
Improvement in NYHA class, *n* (%)	47 (66.2%)	17 (63.0%)	30 (68.2%)	0.652
Improvement in EF > 10%, *n* (%)	24 (29.7%)	9 (29.0%)	15 (30.0%)	0.926
**Postoperative complication**				
Device implantation due to AV block III, *n* (%)	15 (15.0%)	3 (7.1%)	12 (21.4%)	0.051
Left bundle branch block, *n* (%)	15 (15.0%)	5 (11.6%)	10 (18.2%)	0.572
**Paravalvular aortic regurgitation**				
grade I–II grade II grade III	10 (10.0%) 2 (2.0%) -	4 (9.1%) 0 (0%) -	6 (10.7%) 2 (3.6%) -	0.788 0.311
Acute kidney injury, *n* (%)	11 (11.0%)	5 (11.6%)	6 (10.7%)	0.886
Bleeding, *n* (%)	5 (5.0%)	2 (4.8%)	3 (5.5%)	>0.99
Delir, *n* (%)	4 (4.0%)	1 (2.3%)	3 (5.5%)	0.631
Stroke, *n* (%)	2 (2.0%)	1 (2.3%)	1 (1.8%)	>0.99
Length of stay in Intensive care unit (days)	2.0 (1.5–14.0)	2.0 (2.0–20.0)	1.0 (5.0–6.0)	0.554
Length of stay in hospital (days)	13.0 (9.0–17.0)	13.0 (11.0–17.0)	11.5 (9.0–17.0)	0.180
Rehospitalization within 30 days, *n* (%)	10 (10.2%)	3 (7.1%)	7 (12.5%)	0.509
Worsening eGFR > 10% within 12 months	32 (42.1%)	15 (51.7%)	17 (36.2%)	0.182
**Mortality**				
In-hospital mortality, *n* (%)	5 (5.0%)	3 (7.0%)	2 (3.6%)	0.650
6-month mortality, *n* (%)	5 (5.0%)	3 (7.0%)	2 (3.6%)	0.650
12-month mortality, *n* (%)	6 (6.0%)	3 (7.0%)	3 (5.4%)	>0.99

**Table 3 nutrients-14-00203-t003:** Sex-specific correlation of NEFA concentrations before and after TAVI with physical performance over 12 months.

	Female		Male	
	Spearman Rho	*p*-Value	Spearman Rho	*p*-Value
**Before TAVI**				
NEFA with 6-MWT	−0.552	0.001	−0.007	0.964
NEFA with bicycle riding	0.218	0.274	0.103	0.490
**6 months after TAVI**				
NEFA with 6-MWT	0.278	0.169	0.077	0.660
NEFA with bicycle riding	−0.223	0.306	−0.039	0.839
**12 months after TAVI**				
NEFA with 6-MWT	−0.167	0.456	−0.062	0.764
NEFA with bicycle riding	0.056	0.803	−0.190	0.342

## Data Availability

Not applicable.

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
