# Peer review of "Physical Performance and Non-Esterified Fatty Acids in Men and Women after Transcatheter Aortic Valve Implantation (TAVI)"

_nutrients, 2022, doi:10.3390/nu14010203_

Round 1

Reviewer 1 Report

This is an interesting observational study investigating NEFA in a cohort of 100 patients undergoing TAVI. The design seems appropriate and the methods are largely well described.

My questions relate mainly to the interpretation and generalisability of the results:

1) The study flow describes 345 patients meeting exclusion criteria – what were these?

2) The m:f proportion is roughly balanced (44:56). Is this typical in the authors’ centre or were selection measures taken to preserve this balance? A male predominance is typical for most TAVI series.

3) The main findings are of elevated NEFA concentrations in both men and women at baseline, which after TAVI regress into the RR for men but not women. As there is no control cohort, the authors should provide greater detail on how these reference ranges have been derived and whether they are applicable to the study population. The references currently provided seem to be [19] a German biochemistry textbook and [20] a study in 26 healthy subjects (mean age 29 years).

4) The authors note an inverse correlation between NEFA and 6MWT distance at baseline in women but not men. What is the purported mechanism for this?

5) TAVI was associated with an increase in 6MWTD in men but not women. The authors speculate that this may be related to increased NEFA uptake in skeletal muscle. Diminished improvement in 6MWTD after TAVI in women has been noted previously [Circulation. 2017;136:632–643], I think the present manuscript would be strengthened by discussion of these findings.

6) Median baseline 6MWTD in women is stated as 197 (131 – 330) m in the text but looks higher than this in the box plot in Figure 3 (c. 220m)? Could the authors clarify?

7) Regardless, the sex differences in baseline 6MWTD point toward the included women being relatively more deconditioned, which would likely have implications for their functional improvement after TAVI. Could the authors comment on this?

8) Are these apparent sex differences in change in 6MWT after TAVI still evident after adjustment for baseline capacity? Table 3 suggests there was not difference in the proportion of women vs men increasing 6MWTD by more than 10%.

9) We know that frailty is a critical determinant of functional change after TAVI (e.g. JACC:CI 2019:781-9), how was frailty assessed in the present study?

10) Time spent bicycling is included as a functional outcome. This is translatable and interesting but I’m not sure how applicable it is to a TAVI population. For instance, how many patients (particularly women) in the present study were bicycling at the start of the study – looks like n=3 (Figure 3b)?

Author Response

Manuscript ID nutrients-1500468

„Physical performance and non-esterified fatty acids in men and women after transcatheter aortic valve implantation (TAVI)“

This is an interesting observational study investigating NEFA in a cohort of 100 patients undergoing TAVI. The design seems appropriate, and the methods are largely well described. My questions relate mainly to the interpretation and generalisability of the results:

  • The study flow describes 345 patients meeting exclusion criteria – what were these?

AUTHORS RESPONSE

We thank the reviewer for this comment and have added a sentence in the methods section of the revised manuscript. Exclusion criteria were lack of legal capacity, life expectancy < 1 year, and missing baseline blood draw.

  • The m: f proportion is roughly balanced (44:56). Is this typical in the authors’ center or were selection measures taken to preserve this balance? A male predominance is typical for most TAVI series.

AUTHORS RESPONSE

The Brandenburg Heart Center provides supra-regional care for patients requiring TAVI. Gender distribution in TAVI patients of this study is similar compared to the total cohort of our center during the last years. In a recent retrospective study, we reported comparable proportions between men and women in a cohort of 800 TAVI patients. (Clin Kidney J. 2020 Nov 3;14(1):261-268. doi: 10.1093/ckj/sfaa179).

  • The main findings are of elevated NEFA concentrations in both men and women at baseline, which after TAVI regress into the RR for men but not women. As there is no control cohort, the authors should provide greater detail on how these reference ranges have been derived and whether they are applicable to the study population. The references currently provided seem to be [19] a German biochemistry textbook and [20] a study in 26 healthy subjects (mean age 29 years)

AUTHORS RESPONSE

We agree. Sex-specific reference ranges reported in the literature are generated from the normal population and are not adjusted for age. To the best of our knowledge, data reporting sex-specific reference ranges for NEFA in elderly patients or those undergoing TAVI are lacking.

Taking this into consideration, we have added three additional references reporting NEFA concentrations in >10,000 partly elderly patients. In all three studies, the NEFA concentrations ranged within the upper limit of the reference range reported for the normal population.

  • Miedema et al. (Am J Cardiol 2014 Sep 15;114(6):843-8.) analyzed NEFA in 4707

participants from the Cardiovascular Health Study with a mean age of 75-year and 58% being women. The mean concentration of NEFA was 0.41 ± 0.03 mmol/L.

  • Pilz et al. (Eur Heart J. 2007 Nov;28(22):2763-9) reported NEFA concentration in

3315 patients with a median age of 63 years who had undergone coronary angiography. They found elevated plasma NEFA are an independent risk factor for future sudden cardiac death in patients referred to coronary angiography. The median concentration of NEFA in this cohort was 0.53 (0.49-0.58) mmol/L. Of interest, in the highest quartile of NEFA (1.15 mmol/L), the authors report the highest percentage of women (37.1%) compared to the lowest quartile of NEFA (0.34 mmol/L) with 18.3%.

  • Jin et al. (Cardiovasc Diabetol. 2019 Oct 14;18(1):134) reported a median concentration of NEFA of 0.40 (0.30-0.53) mmol/L in 5433 patients with a mean age of 58 and with stable coronary artery disease.

4) The authors note an inverse correlation between NEFA and 6MWT distance at baseline in women but not men. What is the purported mechanism for this?

AUTHORS RESPONSE

The observed inverse correlation between NEFA and 6MWT distance at baseline in women but not men might relate to a higher BMI, a slightly higher age in women compared to men, and more severe heart failure (higher proportion of NYHA IV) in women. We have added a corresponding sentence to the discussion section of the revised manuscript.

  • TAVI was associated with an increase in 6MWTD in men but not women. The authors speculate that this may be related to increased NEFA uptake in skeletal muscle. Diminished improvement in 6MWTD after TAVI in women has been noted previously [ 2017;136:632–643], I think the present manuscript would be strengthened by discussion of these findings.

AUTHORS RESPONSE

We agree and have added a paragraph of the above-mentioned study regarding female sex being an independent factor for reduced exercise capacity improvement after TAVI measured by 6MWT.

  • Median baseline 6MWTD in women is stated as 197 (131 – 330) m in the text but looks higher than this in the box plot in Figure 3 (c. 220m)? Could the authors clarify?

AUTHORS RESPONSE

We thank the reviewer for this comment and double-checked the results. We recognized that we used list-wise deletion when generating boxplots compared to pair-wise deletion for values presented in the table. Unfortunately, we did not mention this in the initially submitted manuscript. In the revised manuscript, we have added a sentence to the methods section. Also, we have added this information in the table footnote and in the figure legends.

We think that both, list-wise and pair-wise deletions make sense where appropriate. The boxplot shows 6MWT results over time. Therefore, we used list-wise deletion in the boxplot.

  • Regardless, the sex differences in baseline 6MWTD point toward the included women being relatively more deconditioned, which would likely have implications for their functional improvement after TAVI. Could the authors comment on this?

AUTHORS RESPONSE

We agree with the reviewer that the female patients in our study might be more deconditioned compared with the male patients before TAVI and that the lack of improvement in 6MWT in females might reflect a higher proportion of females with more severe comorbidities, such as more frequent NYHA class IV and BMI outside the normal range.

In addition, we did not assess preexisting orthopedic conditions or psychiatric diseases such as depression in the study, both comorbidities may be more prevalent in elderly women and could affect the performance of the 6MWT.

Because the primary question of the study was not designed to examine consequences of lower physical improvement, we would be cautious about the clinical consequences/ implications of this study finding.

However, in future prospective studies, systematically implementing exercise capacity assessment pre- and post-TAVI might help to improve patient risk stratification.

8) Are these apparent sex differences in change in 6MWT after TAVI still evident after adjustment for baseline capacity? Table 3 suggests there was no difference in the proportion of women vs men increasing 6MWTD by more than 10%.

AUTHORS RESPONSE

We agree with this reviewer that results regarding 6MWT in Table 2 and Figure 3a appear to be conflicting. Figure 3 shows that 6MWT measured at 3 timepoints in men improved over time after TAVI (Friedman test). However, 6MWT measured at 3 timepoints in women did not improve over time.

We understand the reviewers’ suggestion to compare delta 6MWT as adjustment for baseline capacity (i.e. 6 months or 12 months minus baseline) in men versus delta 6MWT in women. However, such an approach would lose power (several 2 group comparisons) and would potentially require adjustment of p-values for multiple testing.

As this was a pilot study, we would like to suggest not to use such an approach. We discuss the study results as hypothesis-generating findings in the revised manuscript, e.g. in the Conclusions section: “… In this hypothesis-generating study, in men but not women, NEFA concentrations appear to improve into reference range after TAVI. …”. 

9) We know that frailty is a critical determinant of functional change after TAVI (e.g., JACC:CI 2019:781-9), how was frailty assessed in the present study?

AUTHORS RESPONSE

Unfortunately, we did not assess frailty in the present study. We have acknowledged this limitation in the discussion section of the revised manuscript.

10) Time spent bicycling is included as a functional outcome. This is translatable and interesting but I’m not sure how applicable it is to a TAVI population. For instance, how many patients (particularly women) in the present study were bicycling at the start of the study – looks like n=3 (Figure 3b)?

AUTHORS RESPONSE

We thank this reviewer for this comment. Indeed, fewer women spent time bicycling before TAVI compared to men. However, in women, the number of women and the hours per week spent on bicycling remained the same, whereas in men it seemed to increase. We mention the limitation of self-reported physical activity in the Discussion section of the revised manuscript.

Reviewer 2 Report

Dear Aurhor,

Congratulations on your work on NEFA concetrations in patients treated with TAVI. The work is generally well written with clear figures and good overall discussion. I have only two small remarks:

  1. You focus on NEFA concentrations and their correlation with phisycal activity, but I believe that you lack data concerning clinical endpoints, vide VARC-3 criteria. For example I don’t see any information concerning periprocedural vascular complications or moderate/severe paravalvular leak which are known to have impact on clinical outcomes and mortality. See for example PMID: 34076883
  2. The reported hours of bicycle riding seem ale little too arbitrary, and therefore you shouldn’t rely on this as a reliable marker of physical activity. I would also skip the figure concerning this issue.

Best Regards,

Author Response

Manuscript ID nutrients-1500468

„Physical performance and non-esterified fatty acids in men and women after transcatheter aortic valve implantation (TAVI)“

Dear Author,

Congratulations on your work on NEFA concentrations in patients treated with TAVI. The work is generally well written with clear figures and a good overall discussion. I have only two small remarks:

  1. You focus on NEFA concentrations and their correlation with physical activity, but I believe that you lack data concerning clinical endpoints, vide VARC-3 criteria. For example, I don’t see any information concerning periprocedural vascular complications or moderate/severe paravalvular leak which are known to have an impact on clinical outcomes and mortality. See for example PMID: 34076883

AUTHORS RESPONSE

We thank the reviewer for this important remark. Overall, 10 patients (6 male/4 female) developed paravalvular aortic regurgitation grade I-II, 2 patients (both male) grade II, and no patient grade III. Perioperative vascular complications occurred in 4 patients (all male). We have added this finding in the result section of the revised manuscript.

  1. The reported hours of bicycle riding seem ale little too arbitrary, and therefore you shouldn’t rely on this as a reliable marker of physical activity. I would also skip the figure concerning this issue.

AUTHORS RESPONSE

We agree with this reviewer that self-reported bicycle riding is a subjective study endpoint.  We mention the limitation of self-reported physical activity in the Discussion section of the revised manuscript and would like to suggest maintaining the Figure in the manuscript.
